# Microbial Community Composition of the Antarctic Ecosystems: Review of the Bacteria, Fungi, and Archaea Identified through an NGS-Based Metagenomics Approach

**DOI:** 10.3390/life12060916

**Published:** 2022-06-18

**Authors:** Vesselin V. Doytchinov, Svetoslav G. Dimov

**Affiliations:** Faculty of Biology, Sofia University “St. Kliment Ohridski”, 1164 Sofia, Bulgaria; v.doichinov@uni-sofia.bg

**Keywords:** Antarctica, environmental genetics, metagenomics, microbiology, next-generation sequencing (NGS), high-throughput sequencing (HTS), bacteria, archaea, fungi

## Abstract

Antarctica represents a unique environment, both due to the extreme meteorological and geological conditions that govern it and the relative isolation from human influences that have kept its environment largely undisturbed. However, recent trends in climate change dictate an unavoidable change in the global biodiversity as a whole, and pristine environments, such as Antarctica, allow us to study and monitor more closely the effects of the human impact. Additionally, due to its inaccessibility, Antarctica contains a plethora of yet uncultured and unidentified microorganisms with great potential for useful biological activities and production of metabolites, such as novel antibiotics, proteins, pigments, etc. In recent years, amplicon-based next-generation sequencing (NGS) has allowed for a fast and thorough examination of microbial communities to accelerate the efforts of unknown species identification. For these reasons, in this review, we present an overview of the archaea, bacteria, and fungi present on the Antarctic continent and the surrounding area (maritime Antarctica, sub-Antarctica, Southern Sea, etc.) that have recently been identified using amplicon-based NGS methods.

## 1. Application of the Next-Generation Sequencing (NGS) Technology for Analysis of Antarctic Samples

### 1.1. Beginning of the Antarctic Environmental Biology

Before the mid-1980s, microbiological taxonomy relied heavily on biochemical, physiological, and morphological criteria for species identification [1]. The evolutionary significance of these attributes was largely arbitrary and based solely on what microbiologists were able to observe at the time. The development of reverse transcription sequencing [2] and the polymerase chain reaction [3] opened the door to using bacteria’s innate DNA sequence to establish a more rigid phylogenic connection between organisms. This paradigm shift toward molecular techniques demanded a revision of the taxa classified at the time and established DNA sequencing techniques as the standard for future species identification and phylogenetic grouping. Before the development of these methods, many of the genera reported from the Antarctic habitats had changed in concept [1]. The first DNA-based approaches for categorizing Antarctic organisms used 5S rRNA to identify prokaryotes in rock samples from the Dry Valley [4] and 16S sequencing of the Antarctic strains from the Deep Lake [5]. Today, the DNA sequence of the 16S rRNA molecule of the small ribosomal subunit is most commonly used to identify bacteria and archaea. The 18S rRNA, ITS1, and ITS2 genomic sequences serve the same purpose for eukaryotes, such as fungi.

### 1.2. The Rise of Environmental Metagenomics

Starting just before the 2010s, a general shift toward NGS was observed, not only for the Antarctic ecosystems but for most global research on environmental diversity. This movement was catalyzed by the advancement of massively parallel high-throughput sequencing platforms, such as Illumina (Solexa), Ion Torrent, Roche 454, SOLiD, and others. These technologies allowed for the fast and cost-effective cataloging of large and various amounts of DNA through the simultaneous sequencing of millions of short fragments (~150–300 bp). Thus, metagenomics, sometimes referred to as DNA metabarcoding, began as an approach for quick annotation of the microorganisms present in an environmental sample with fairly reproducible results, although primer bias is a known issue for amplicon sequencing. A drawback of these technologies is the short read length, which could lead to inaccuracies when extrapolating the host species from amplicon sequence data alone, but recent developments in more sophisticated platforms from PacBio and Oxford Nanopore address these issues. However, the latter platforms pose a more expensive approach and, in some cases, a lower per-base read accuracy [6].

There are generally two approaches to classifying the microbial population of a given sample—amplicon and shotgun-based sequencing (Table 1). In amplicon sequencing, a specific reporter DNA fragment (amplicon) is amplified, sequenced, and used for taxonomic identification. In shotgun-based methods, the genetic information of the sample is fragmented, and each piece is sequenced, following stitching of the reads, based on their overlap, into a comprehensive picture of the community. While the shotgun-based approach offers an in-depth profile of the community down to the species and gives data on the genes and biochemistry of the microorganism, it is also accompanied by a tremendous bioinformatical challenge because an environmental sample contains millions of microorganisms, each with its own unique genome, millions of bases long. The computation needed for stitching such a large array of fragments together, aligning all that data whenever you need to access such a database, or simply storing all of that information, pose a challenge.

There are two workflows for environmental biodiversity assays, i.e., culture-dependent and culture-independent. In culture-dependent analysis, the samples are plated and grown in laboratory conditions to enrich the number of microorganisms present before sequencing is carried out on the desired clones. This cultivation step is omitted in culture-independent methodologies, where DNA is directly extracted from a sample. One of the most significant benefits of using culture-independent DNA sequencing to identify organisms is detecting species that are difficult to cultivate in a laboratory. However, this usually leads to much more significant and often different biodiversity than the one caught using culturing techniques [7].

### 1.3. 16S Hypervariable Regions

The short fragment reads of NGS (~150–300 bp) limit only specific regions of the 16S rRNA gene (roughly 1.5 kb in length) to be sequenced in a single run. Compared to sequencing the entire 16S gene, this is still a viable option because the gene contains both hypervariable and conserved regions (Figure 1a). The reason for the existence of these regions is the same reason the 16S molecule is used for taxonomic classification, that is, the fact that all life requires protein and the ribosomes that synthesize them. The conserved regions of the 16S gene are simply those in which changes to the nucleotide sequence would render the ribosome non-functional, and thus, the cell would not survive. The conserved regions also allow the creation of universal primers capable of complementing the 16S rRNA gene of most organisms. However, it has been well documented that the chosen sequence region significantly influences the results of a particular environmental survey [7]. The V3 and V4 variable regions are the most commonly sequenced around various habitats. Studies focused on the different 16S regions could be performed to picture a more comprehensive review of NGS sequencing in Antarctica. Still, caution should be exercised whenever cross-region comparisons are drawn.

Traditionally, 16S rRNA similarity of 97% is used as the threshold for operational taxonomic unit (OTU) delineation, roughly translating to 70% of overall DNA similarity. Values above 97% have been frequently suggested and used in some studies [8] or region-specific cutoff values [7]. Amplicon sequence variants (ASV) represent a newer method for grouping the sequences compiled from metagenomic studies and boast higher precision, distinguishing between single nucleotide changes [9]. ASVs are also referred to as exact sequence variant (ESV) or zero-radius OTU (zOTU).

### 1.4. Richness and Diversity Indices

Community richness is a measure that considers the number of different species (or OTUs) in a sample. In statistical analysis, it is most commonly reflected by the Chao1 and ACE indices. On the other hand, diversity is a measure that considers the relative abundance of the different species in a sample (or sequences in the different OTUs) and is reflected by the Shannon and Simpson indices. Community composition or structure is used in this review to refer to the specific profile of present species and their relative abundance.

Alpha diversity refers to the diversity or richness within a single sample and is typically represented by the Shannon, Simpson, Chao1, ACE, PD whole tree, and Good’s coverage indices. The Shannon index is usually within the range of 1.5–4 and directly represents diversity, assuming that all species are present and randomly distributed in the sample. While similar, the Simpson index gives more weight to the dominant species in a sample and can be thought of as the likelihood of two randomly picked individuals being the same species. PD whole tree gives a measure of diversity based on sample phylogeny. The Chao1 and the ACE indices reflect how representative the sample is to the whole of the community and are assessed based on the number of singletons and doubletons within a sample. In a well-representative sample, these two indices match the total number of discovered OTUs. Similarly, Good’s coverage also reflects how representative a sample is, but the values of this index are interpreted differently. For example, a Good’s coverage of 1 represents a sample with no singletons. At the same time, 0.5 would be a sample where half of OTUs only have a single sequence assigned to them, which would point to an insufficient sampling strategy. 

Beta diversity is the comparison between the diversity of at least two samples. It is typically represented through the UniFrac distance, where a value of 0 equates to no difference in the communities. UniFrac can be further divided into weighted UniFrac distance, where taxa abundance is taken into account, and unweighted, where only the absolute presence of a specific taxon is considered.

If we summarize, the amplicon-based metagenomic studies have the advantage of being less expensive but sufficiently informative. However, some shortcomings could not be neglected. The major one is that they rely on PCR amplification, during which some less-represented DNA matrices could be lost. This shortcoming could be overcome by a whole-metagenome sequencing, which does not include the PCR amplification step. Still, it comes at a much greater price and usually requires a greater amount of DNA, which is not very easy to obtain from the specific Antarctic samples (for example, freshwater or air). Another shortcoming is that Antarctic microbiota is particularly specific, and many of its representative species are not present in the DNA sequences databases, thus making the analyses very difficult or impossible. Finally, the third major shortcoming is of methodological nature; as Antarctica is quite remote and isolated, and the laboratory infrastructure is rather scarce, the samples or the DNA isolated from the samples must be preserved for more extended periods and transported at great distances. This long-term conservation and transportation could also affect the quality of the DNA.

## 2. Community Composition of the Antarctic Ecosystems

Antarctica is the coldest continent in the world, with mean annual temperatures ranging from −5 °C at the Antarctic Peninsula to −50 °C toward the inland plateau and experiencing frequent freeze–thaw cycles [10]. Precipitation, air humidity, and salt concentration are low and depend on the Southern Ocean’s proximity surrounding the continent [10]. A total of 99.6% of the landmass is covered with snow or ice, although seasonal variations are observed [11]. The only parts of the continent that are regularly not covered are scarce coastal areas, nunataks, and mountain peaks. The lack of organic nutrients and liquid water, especially in the inland plateau, is the main limiting factor for life. Additionally, the high solar radiation poses an obstacle to microbial life development on the continent [12]. This explains why biodiversity is most prominent in the coastal regions, especially the Antarctic Peninsula, which has a much more hospitable environment [10,11].

The main interest in studying the Antarctic microorganisms comes from their enormous potential for isolating new biologically active and valuable products. This potential is due to the unique and harsh environment, which caused the selection of strains and species showing unusual metabolic properties and/or the production of unusual metabolites and substances. Substances possessing antimicrobial activities have been reported both for bacteria [13,14] and fungi [15]. The UV radiation, the low temperatures, and the constant freezing and thawing cycles make Antarctic bacteria ideal candidates for the isolation of pigments acting as extracellular sunscreens or for use as natural biodegradable pigments and colorants [14]. The same extreme conditions caused the evolution of fungal strains that possess cold-adapted enzymes and compounds such as amylases, cellulases, chitinases β-galactosidases, invertases, lactases, lipases, proteases, xylanases, superoxide dismutases, etc., useful in different industries and biotechnological productions [16]. Antarctic fungi have also been reported as a source of novel antibiotics, antifungals, antivirals, and antiparasitic substances [15].

### 2.1. General Trends in Antarctic Microorganism Distribution and Diversity

All Antarctic life, whether plant, animal, or microscopic, is subject to extreme environmental conditions that inevitably steer the evolution and adaptation of species, as they colonize the few available environmental niches. This leads to reduced species diversity and an ecosystem with a simplified trophic structure. However, a strong argument has been made that the generally low biodiversity and the specific species adaptations are better explained by the lack of predictability in the climate of Antarctica rather than its severity [17]. 

Microbial communities in Antarctica can be divided into several groups according to the ecological recesses they occupy—soil and rock communities, inland and glacial water communities, meltwater ponds communities, subglacial water communities, marine water communities, microbial mats and sediments communities, cryoconite holes, glacial ice, and snow communities, and airborne microorganisms’ communities (Figure 2). They will be discussed separately further in this review.

In terrestrial Antarctic environments, this unpredictability comes from sudden freezing, even in the summer, which dictates a relatively fast pace of living when defrosted, since resources are generally not restricted. On the other hand, marine environments below the depth of 2 m typically do not freeze but face additional challenges, such as the seasonality of food, which is coupled with the blooming of phytoplankton. This seasonality requires a slower pace of life but allows for extended overall biodiversity due to the lack of sudden environmental changes to dictate a narrower set of adaptations. Indeed, some Antarctic marine environments have been found to have greater biological diversity than global averages [17].

Lake ice sediment microorganisms have also been identified in surrounding cyanobacterial microbial mats at a distance as far as 15 km away [18]. This shows the excellent dispersal force of Antarctica’s katabatic winds and suggests that microorganisms opportunistically colonize the lake ice sediment. Similarly, the cyanobacteria inhabiting cryoconite holes in Antarctica’s McMurdo Dry Valley region have been found in a separate study of lake ice and microbial mats from the same area [19]. In short, Antarctic microorganisms seldom inhabit a single habitat on the continent unless they are faced with unusually high isolation, such as that of subglacial lakes or fossilized sediments.

An inverse correlation between Antarctic bacterial diversity and density and latitude has been noted for soils without vegetative cover [20,21]. It is also documented for the native fauna and flora [22,23]. However, several studies, including ones conducted in other continents, have found no such trend [24,25] or have attributed the trend to other factors, such as soil pH [26]. Additional studies on Antarctica that describe the bacterial diversity of vegetation-covered and exposed soils and accounting for soil factors, such as pH, would be needed to elucidate the connection with latitude.

### 2.2. Sources of Microbial Life

Biological matter can reach Antarctica from South America and other nearby Antarctic locations, contributing to the species diversity [27,28]. Native plants and mosses naturally act as ‘hotspots’ for microorganism diversity by contributing to an increase in available organic matter, moisture, nitrogen, and carbon in the soil, as well as sheltering microorganisms from the selective pressure of the low Antarctic temperatures [21,29]. On the other hand, interactions with the limited number of plant species inevitably lead to a different selective force acting on those microorganisms [30,31,32]. The activity of native fauna similarly contributes to microorganism diversity, most notably from birds’ nesting sites, which typically enrich the underlying soil with various organic matter (guano, feathers, eggshells, etc.) [29]. Indeed, penguin guano is one of the most prominent sources of organic matter in Antarctic soils [33]. It has been implicated in affecting the bacterial composition of local plant rhizospheres [34] by increasing the number of *Firmicutes*, since they are the dominant phylum found in penguin guano and ornithogenic soil [35,36].

### 2.3. Terrestrial Communities

Most of the microbial community research on the Antarctic continent concerns soil and rock samples. Thus, significant knowledge has been accumulated about these environments. The most prominent conclusion from this research is the dispelling of the pre-20th century assumption that an isolated and inhospitable continent, such as Antarctica, would have negligible, if any, microbial diversity. However, the soils and environments of the South Pole proved to be far from sterile, hosting a variety of microbial communities. This represents a challenge for generalizing the microbial communities of Antarctic soils, since their composition can vary greatly, even between small distances. For example, the McMurdo Dry Valleys, which are one of the most commonly sampled locations, constitute roughly 4000 km^2^ and comprise areas with varying altitudes and environmental conditions, such as moisture, vegetation, exposure to UV radiation, etc., and hosting a plethora of communities.

#### 2.3.1. Relationship between Geography and Microorganism Communities in Antarctic Soils

Soils from Antarctica contain *Acidobacteria*, *Actinobacter*, *Proteobacteria* (most commonly *Alpha*- and *Gammaproteobacteria*), *Bacteroidetes*, and *Firmicutes* as the dominating phyla but in different relative abundance, depending on both the microenvironment of the site and its broader geographical location [37,38,39,40,41,42]. The phyla *Planctomycetes*, *Verrucomicrobia*, *Chloroflexi*, and *Cyanobacteria* are less abundant but still commonly found in soil surveys. 

The diversity of fungi on the continent is much more limited, with the dominating phylum always being *Ascomycota* at more than 70%, followed by *Basidiomycota* and less than 10% for any other phyla [40,43,44]. It is speculated that this low diversity could be a result of the specific geology of the area, the larger and harder to transport spores of fungi, or the lack of vegetation to serve as vectors for dispersal [40].

A metastudy of 13 Antarctic soil surveys showed a weak correlation between geographical location and bacterial community composition [45]. Furthermore, sites just 3 km apart exhibited varying taxonomy. No latitudinal effect was present, but community composition could be clustered into two groups, namely the Antarctic Peninsula and Victoria Land, respectively, owing to members of *Actinobacteria*, *Acidobacteria*, and *Bacteroidetes*. The same study also pointed to the ocean as bacterial inoculum for the continent and highlighted the effect nitrogen, moisture, pH, and conductivity had on community composition.

A study of soil from Terra Nova Bay revealed that soils from only 4 km apart exhibit great variety in bacterial phyla abundances [46]. However, pH, water content, conductivity (salt), SiO2, soil texture, and heavy metals (Pb, Cu) greatly impacted community composition. Thus, the authors concluded that the environmental variables have a more significant effect than the spatial distance. Similarly, varying community composition was noted for soils of the McMurdo Dry Valleys at sites only 2 km apart [47]. Another study of various points within the McMurdo Dry Valleys noted that only 2 of the approximately 200 OTUs were found in all samples, again pointing to either minimal inter-valley redistribution or the result of highly selective pressures of the physicochemical characteristics of the areas [38].

Due to its lower altitude and proximity to the Southern Ocean, the Antarctic Peninsula and the islands of maritime Antarctica are the most hospitable to life and offer a hotspot for scientific research. Biodiversity ‘hotspots’ have also been recorded at several continental Antarctic locations. In a study of Mitchell Peninsula soils, a high diversity of bacteria and fungi has been documented [44]. Three phyla of bacteria dominated the site—*Chloroflexi*, *Actinobacteria*, and *Proteobacteria*, with the most abundant class being *Thermoleophila*. *Ascomycota* dominated the fungal sequences and was represented mainly by the orders *Teloschistales* and *Lecanorales* from class *Lecanoromycetes*. A high number of bacterial and fungal candidate divisions were also discovered, with many of the dominant classes being poorly characterized. Interestingly, longitude was the best determining factor for community composition, and differences in diversity were noted at higher elevations. The same study highlighted a correlation in the community composition of bacteria and lichenous fungi but did not detect a significant amount of *Cyanobacteria*, which points to phototrophic *Chloroflexi* or algae taking their place as primary producers.

Mars Oasis, Alexander Island, can also be considered a hotspot with increased bacterial, fungal, algae, and protist diversity [20,24], correlating with increased diversity of fauna in that area [48,49]. Soil samples from Fossil Buff and Coal Nunatak in Mars Oasis are dominated by pink-pigmented *Methylobacterium* spp. [20], which could be attributed to selective environmental pressure, such as intense UV light, scarce water, and low amounts of nutrients. Indeed, *Methylobacterium* spp. use C1 compounds as a facultative energy source [50] and have been documented as being UV resistant [51]. A more recent metagenomic study confirmed the high degree of biodiversity, reporting *Proteobacteria* (mostly *Alphaproteobacteria*), *Actinobacteria*, *Firmicutes*, and *Bacteroidetes* dominating the soils [39]. Fungi were not as highly represented, the detected species being *Gibberella* sp., *Neurospora* sp., *Magnaporthe* sp., *Schizosaccharomyces* sp., *Saccharomyces* sp., and *Eremothecium* sp.

The freezing climate of Antarctica allows for the existence of extremely sharp temperature gradients on the continent. Such is the case on Deception Island, Antarctic Peninsula, where an active volcano gives rise to cracks in the earth’s surface called fumaroles, which emit hot steam and gasses. A study comparing the communities of local glaciers and fumaroles soil found that members of three phyla are common to both extreme environments—*Proteobacteria* (mainly *Gamma*- and *Alphaproteobacteria*), *Planctomycetes* (*Planctomycetacia*, *Pirellulales*), and *Bacteroidetes* (*Bacteroidia*, *Flavobacteriales*), with the most abundant phyla in the fumaroles soil being *Calditrichaeota* and *Chloroflexi* [52]. Interestingly, a separate study of microbial mats in the same area and sampling different temperature sites from 88 °C to 2 °C found a similar phylum level composition along the temperature gradient but with much higher bacterial diversity at the lower temperatures [53]. *Proteobacteria* and *Deinococcus-Thermus* were most abundant in the high temperature ranges but with the addition of *Cyanobacteria*, which dominated both extremely hot and cold samples. More specifically, a higher proportion of the orders *Flavobacteriales* and *Cytophagale* was observed, as well as the fungal order *Rhizophydiales,* belonging to the only identified fungal phylum *Ascomycota*. It was also shown that heat-tolerant bacteria from orders *Kallotenuales*, *Bacillales*, *Thermales*, *Chthonomonadales* and Gp16, and the nitrate-oxidizing bacteria from order *Nitrospirales*, correlated with each other more strongly in the hottest mat.

#### 2.3.2. Relationships between Edaphic Factors and Microorganism Communities in Antarctic Soils

On a local scale, edaphic factors, such as pH, nitrogen, carbon, water, conductivity, temperature, heavy metals, and others, have arguably the most significant impact on the microbial community composition of the Antarctic soils [45,46,47,52,54].

The McMurdo Dry Valleys represent the largest ice-free zone on the Antarctic continent [55,56] and thus a strong focus for microbiological research. However, soils of this region contain much less organic matter than regular soils, which leads to more oligotrophic species dominating the area, such as *Acidobacteria* [54]. The effects of edaphic factors on soil microorganisms can be differentiated when comparing local and regional sites. This was shown in a soil study of the Taylor and Wright valleys, both of which are a part of the McMurdo Dry Valleys [54]. More specifically, pH, organic matter, and moisture positively impacted biomass and diversity, while sulfate had a noticeably negative correlation. The authors concluded that snow patches most likely serve as ‘resource islands’, offering a steady moisture supply to the soil. However, community composition did not vary between the exposed and snow-covered soil, but it did vary between the two valleys.

Soils at lower elevations, containing increased organic carbon and moisture, are typically dominated by *Acidobacteria* and *Actinobacter*, while highly elevated areas with lower moisture and organic carbon areas contain more *Firmicutes* and *Proteobacteria* [54]. However, a separate study shows that total carbon correlates negatively with *Acidobacteria*, which appeared more common at neutral pH [37]. Low salinity also favors the phyla *Acidobacteria* and *Actinobacteria*, while high salinity promotes an abundance of *Firmicutes* [47]. The dominance of *Firmicutes* in harsh conditions is most likely due to their thick Gram-positive cell walls and sporulating ability, which protects them from desiccation and helps long-term survival [57,58].

Studies of soil from frost boils, which represent rocky patches in the vegetation, revealed that eastern Antarctic soils in the area of Browning Peninsula are dominated by *Actinobacteria*, followed by *Chloroflexi* [40,43]. In the same studies, it was revealed that fungi of the phylum *Ascomycota* represented more than 95% of the identified fungal species, followed by *Basidiomycota*. Interestingly, chlorine, phosphate, and elevation impacted bacterial community composition but not fungal. Authors suggested that the lower fungi diversity could result from the boils in the area, the larger and harder to transport spores of fungi, or the lack of vegetation to serve as vectors for dispersal [40]. Greater local homogeny in community composition was also shown for areas with frost boils than less patchy connected soils [43]. The authors explain this by the freeze–thaw cycles, which create a barrier and homogenize the soil. A survey of Windmill Island in the Browning Peninsula showed that pH is the primary driver of community composition but also found that soil fertility, described as organic matter, nitrogen, and chloride, is the driving force behind microbial richness [59]. Authors suggest that the pH establishes the initial community composition, members of which begin to dominate based on the fertility of the soils. The same study also pointed out that phosphorus is the next critical factor determining richness and diversity.

The presence of *Cyanobacteria* within Antarctica’s soils is highly dependent on high levels of moisture and conductivity but not so much on pH [46,60]. *Cyanobacteria* are not one of the dominant phyla within Antarctica’s soil communities, generally being absent in some cases [38,54]. This could be due to competitive pressures, since a study in Dronning Maud Land, east Antarctica, identified diverse microorganisms capable of assimilating CO_2_ that are not *Cyanobacteria* [61].

A study of soil near lake Fryxell, McMurdo Dry Valleys, provided key insights into the effects that drastic changes in the water content and the organic matter in soils could have on community composition [47]. Researchers found that adding resources to the soil increased respiration but not for soils with high salinity. For low- and medium-salinity sites, the influx of water and organic matter led to increased *Proteobacteria* and *Firmicutes*. Still, high-salinity sites were strictly dominated by *Firmicutes*, primarily *Paenisporosarcina* sp. Thus, the authors concluded that Antarctica’s soil ecosystems have little resistance to environmental change. A similar study, in which temperature was gradually increased by 2 °C over a three-year period at three different islands of the Antarctic Peninsula and maritime Antarctica, including both vegetation-covered and pristine soils, showed that the microbial community exhibited a rapid response to the changing environment [42]. Still, community composition was most strongly affected by edaphic factors, location, and the vegetative state of the soil. The shift in the major taxa included decreased *Acidobacteria*, while *Alphaproteobacteria* increased during the three-year period. Although bacterial and fungal richness was higher, diversity remained the same but with a different microbial profile. An increase in fungi points to increased soil respiration and long-term carbon sequestering due to the high carbon assimilation by fungi. Authors also noted that bacterial genotypes had become more functionally homogenous, with minor variation between nitrogen and carbon cycling genes, most likely due to a shift from specialized Antarctic species to generalist species.

The archaea in soils showed very limited richness, with an average of six OTUs per sample, but higher values correlate with increased water content [62]. The archaea were dominated by *Thaumarchaeota* affiliated with the Marine Group 1.1b, while the rest belonged to *Euryarchaeota*. *Nitrososphaera gargensis* was the closest relative to all of the *Thaumarchaeota* sequences. Researchers attribute *Thaumarchaeota*’s presence to the resistance of this phylum to freeze–thaw cycles and note their role in the nitrogen cycle as ammonia oxidizers. A survey of soils from fumaroles near a volcano on Deception Island revealed that the main driving factor for archaea community composition was temperature, while for bacteria, this included pH, salinity, sulfate, and nitrogen [52]. In the same study, archaea also dominated the hotter fumaroles, where the dominant genus was the hyperthermophile *Pyrodictium*, while *Euryarchaeota* (specifically *Thermoplasmata*, Marine Group II) were the prevailing archaea in the glacial samples. Nitrogen and methane cycles also appear to have a pivotal role in Antarctic glaciers, where bacteria from the phyla *Verrucomicrobia* and *Patescibacteria* are found in high abundance and cooccur with the possibly methanogenic *Methanomassiliicoccus*. In another soil study in Mars Oasis, the genus *Methanosarcina* was the most abundant [39].

#### 2.3.3. Roles of Different Microorganism Groups in Antarctic Soils

Attributed to the unique ecological niche they occupy, the different bacterial phyla play specific roles in Antarctic soils. In soils with an organic content higher than the normal annual acquisition, such as the soil underneath mummified seal carcasses [63], ornithogenic soils [64], or when adding such organic matter to the soil in a controlled setting [47], the high amount of *Firmicutes* and *Gammaproteobacteria*, but also *Bacteroidetes*, is most apparent. *Firmicutes* represent a group of sporulating bacteria with thick cell walls [57,58], which most likely benefit from long-term survival in the harsh Antarctic environment. It could then be speculated that members of this phylum lie dormant in Antarctic soils for prolonged periods, most likely as spores, and upon the rare influx of organic matter, they rapidly outcompete any autotrophs and become dominant within the community composition. At the same time, some *Gammaproteobacteria* can degrade high and low molecular weight organic carbons [65]. Similarly, some *Bacteroidetes* species have a documented role in the degradation of high molecular weight compounds [66]. Overall, it is clear that these three groups play an essential role in the mineralization processes of the Antarctic soils [64]. 

A study on the impact of edaphic factors in Liivingston Island, maritime Antarctic, proposed that the organic compounds and the lower pH provided by the bryophytes (primarily mosses) contributed to the dominance of *Bacteroidetes* in vegetation-covered soils compared to mineral soils [37]. This points to the crucial role *Bacteroidetes* might have in the mineralization of vegetative soils, while in the same study, *Acidobacteria* were negatively correlated with mineralization.

#### 2.3.4. Lithobiont (Rock) Communities

Hypolithic communities are crucial to the carbon and nitrogen cycling of desert ecosystems by protecting microorganisms from extreme environmental factors, such as the temperature, the freeze–thaw cycles, the wind, the UV radiation, and additionally by trapping moisture, thus preventing desiccation [67,68,69]. Rocks, in general, provide a stable substrate for the construction of exopolysaccharide matrices that create a stable film of microorganisms on the surface.

In a joint survey of the soil and lithobiont communities in the McMurdo Dry Valleys, *Actinobacteria* was the most prominent phylum, with about half of the recognized sequences, the rest belonging to *Cyanobacteria*, *Gemmatimonadetes*, and *Verrucomicrobia* [70]. *Actinobacteria* was even more heavily represented in the endolithic community, while *Cyanobacteria* was underrepresented. This observation does not match the global trend of *Cyanobacteria* dominating the desert hypolithic communities, where they function as primary producers through photosynthetic carbon fixation and nitrogen cycling [71,72]. The same survey proposes that members of the genus *Gemmatimonas* (phylum *Gemmatimonadetes*) could have a role in phosphate metabolism, while members of *Aciditerrimonas* (phylum *Actinobacteria*) have a role in reducing iron. In the study, the two most abundant genera of *Cyanobacteria* were *Leptolyngbya* and *Phormidum*, which appeared consistent with older evidence of maritime Antarctic assemblages. Interestingly, most of the observed OTUs in both soil and lithobiont samples were shared, even if the two communities clustered separately regarding their composition. This implies the existence of either a synergetic relationship between these communities or echoes a proposed hypothesis that the lithobiont communities function as a diversity reservoir for terrestrials ones [68]. The authors of the study also propose aeolian transport, and the fact that many of the detected lineages are also found in McMurdo Dry Valleys aquatic habitats, yet are absent from soils in the area, supports the hypothesis that the diversity of *Cyanobacteria* is influenced by nearby water bodies [69].

In a global study of the desert hypolithic *Cyanobacteria* [73,74], it was demonstrated that the phylum constituted 70% of the community composition in quartz crystals from the McMurdo Dry Valleys, but almost no diazotrophic members were found, which is consistent with other studies. Instead, the role of nitrogen fixation was attributed to *Alphaproteobacteria*, belonging to the orders *Burkholderiales*, *Rhizobiales*, and *Rhodospirillales*. This suggests that carbon input from *Cyanobacteria* in deserts is accompanied by nitrogen input from other phyla. It was also noted that the Antarctic deserts were very dissimilar in the community composition to other hyperarid deserts, which is most likely due to strong selective forces and geographical isolation.

An essential trait of the hypolithic communities is their development over time, which could explain the perceived inconsistency in the relative abundance of comprising phyla. A study of the hypoliths in Miers Valley, east Antarctica, elucidated the shift of the community composition and also distinguished three types of hypoliths: Cyano (Type I), Fungal (Type II), and Bryophyle (Type III) [75]. The model describes an initial *Cyanobacteria* and *Proteobacteria* (mostly *Alphaproteobacteria*) community establishing a Type I hypolith, which eventually transits to a Type II and is finally replaced by the more specialized, oligotrophic members of *Acidobacteria* and *Actinobacteria* within the Type III hypolith. This occurs because the members of Type I and II communities are predominantly copiotrophic, meaning they compete successfully with other species only when the resources are abundant.

### 2.4. Inland and Glacial Water Communities

Antarctic meltwater ponds are formed from sea ice, glaciers, or ice shelves during the warmer months of summer and spring from the melting of the surface ice. Among other factors, these freshwater ponds could become highly saline due to mixed sea water, marine aerosols, evaporation, and sublimation [76]. Additionally, even larger Antarctic bodies of water can be covered in a sheath of ice, which reduces mixing with the circulating air. This reduced surface disturbance, coupled with the low temperature and sparse wildlife, could lead to a very pronounced stratification of hydrochemical factors, such as oxygen, salt, and temperature, along the water column, which in turn leads to a stratification of microorganisms that take up a specific niche along the gradient [77,78].

#### 2.4.1. Lake Communities

Generally, the Antarctic lake waters are dominated by *Proteobacteria*, *Bacteroidetes*, and *Actinobacteria*, both in the water column and at the rock–water interface [77,78,79,80]. Less abundant but still commonly found phyla include *Chloroflexi*, *Acidobacteria*, and *Firmicutes*.

The shoreline proximity of a lake is an important indicator of its nutrient availability, and thus bacterial abundance, since inland lakes are typically shallower and have a negligible influx of organic matter from fauna. For example, *Chloroflexi* found in lake water have been strongly associated with penguin colonies, which contribute to the community composition of shoreline lakes through the influx of organic matter [78]. Sea sprays are also a source of organic inoculation for the near-shore lakes, typically resulting in higher salt content and marine microbial species present in the lake.

A large share of the detected lake *Proteobacteria* comprises some ultra-oligotrophs and diazotrophs, specifically species belonging to the families *Bradyrhizobium* and *Burkholderiaceae* [78]. These oligotrophs are much less represented in lakes with higher nutrient availability, generally shoreline lakes. *Betaproteobacteria* are especially prevalent in the upper layers of most of the oligotrophic Antarctic lakes and are represented by genera such as *Polaromonas*, *Polynucleobacter*, *Rhodoferax*, and *Limnohabitans* [77].

The genera *Flavobacterium* and *Pseudarcicella* dominated *Bacteroidetes*, especially in shallow lakes [78]. Since *Flavobacterium* has been implicated as part of microbial mat communities, it stands to reason that sediment-associated bacteria significantly impact lake water community composition in shallower lakes.

Vertical community structure can vary, with no clear species richness and distribution pattern along the water column [77,78]. Some cosmopolitan families, such as *Burkholderiaceae* and *Sporichthyaceae*, have no distinct vertical distribution. However, other families, such as *Xanthomonadaceae* and *Pseudomonadaceae*, are more prevalent in the upper water layers, while *Steroidobacteraceae*, *Xanthobacteraceae*, *Alicyclobacillaceae*, *Thiotrichaceae*, *Desulfobacteraceae*, *Desulfobulbaceae*, *Sphingomonadaceae*, *Marinobacteraceae*, *Caulobacteraceae*, *Gallionellaceae*, and the *Cyanobacteria* families *Leptolyngbyaceae* and *Phormidiaceae* have been detected primarily at the bottom of water bodies. *Cyanobacteria* from the depths of the lakes are most likely associated with benthic mats. Anoxygenic bacteria have also been found at the bottom of the water column, represented by *Chromatiaceae* (a purple sulfur bacteria) and *Anaerolineaceae*. In another study, the chemocline (the separation line between freshwater and water with higher dissolved gasses and solids) has been implicated as an assemblage boundary, with *Actinobacteria* and *Bacteroidetes* dominating the upper layers of water and being replaced by *Gammaproteobacteria* (mainly *Alteromonadaceae* and *Marinobacter*) below the chemocline [77]. *Firmicutes* were also found at and below the oxic–anoxic transition zone. Notably, most of the annotated OTUs in the oxic zone networks were associated with heterotrophic bacteria, some of them being affiliated with hydrogen-oxidizing *Hydrogenophaga*. When comparing different water bodies, the depth of the oxic zone had a significant influence on community composition along the water column. Additionally, between different water bodies, the anoxic zones had much less similarity than the corresponding oxic zones, which were highly variable in structure. 

Antarctic lake communities are pretty susceptible to environmental changes and exhibit a low buffer potential, since their properties are primarily influenced by external factors, such as air currents, sea spray, and the activity of the local fauna. This can be explained by the strong effect geographical proximity has on community composition, indicating the large impact of local factors [78]. 

Perennially ice-covered lakes and their communities are also strongly influenced by seasonal environmental changes. For example, the decrease in available light and lower temperatures, which lead to the formation of a thick ice sheath starting from the top–down of the water column, further reduces the light that reaches the water and also hamper water mixing from the strong katabatic Antarctic winds [80,81,82]. Within a specific lake and at a specific depth, the effect of these seasonal habitat shifts are reflected within the community structure and are represented in a general shift from photoautotrophy to chemolithotrophy [80]. More specifically, *Alphaproteobacteria* annotated OTUs had the highest autumn proliferation, along with some archaeal OTUs, which could be linked to chemolithotrophy and active methanogenesis. Still, the phyla *Actinobacteria* and *Bacteroidetes* remained dominant for most of the year.

Archaea found within the Antarctic lakes are limited to Marine Group I *Crenarchaeota*, *Thermoplasmatales*-related *Euryarchaeotes*, and *Methanomicrobia* [80]. Seasonal blooming of Marine Group II, *Methanomicrobia* and *Methanobacteria*, terrestrial and soil Groups *Crenarchaeota*, and Marine Group I *Crenarchaeota,* was also observed. Inside the sediments and cryoconite holes of Lake Untersee, the phylum *Euryarchaeota* has been identified in both oxic and anoxic zones, represented mainly by *Methanomicrobia*. Still, *Halobacteria* and *Methanobacteria* were also detected [83]. 

A study of the lake rock–water interface found the same dominant phyla as those in the water column, i.e., *Proteobacteria*, *Bacteroidetes*, and *Actinobacteria* but with *Alphaproteobacteria* as the main class of *Proteobacteria*, represented mainly by *Sphingomonas Caulobacter*, *Rhodobacter,* and *Brevundimonas* [79]. *Betaproteobacteria* were represented by *Janthinobacterium*, *Duganella*, *Polaromonas*, *Variovorax*, and *Rhodoferax*, while *Gammaproteobacteria* had the most *Pseudomonas* and *Acinetobacter*. *Bacteroidetes* included the genera *Flavobacterium*, *Pedobacter*, *Prevotella*, *Hymenobacter*, and *Arcicella*. 

The Antarctic continent also harbors about 400 subglacial lakes, the largest being Lake Vostok, which has been covered by ice for about 15 million years [84]. These lakes are entrapped under large glaciers, which serve as a source of water and nutrients, by eroding the bedrock and soil in contact with its surface and releasing particles entrapped in the glacier through ice melting. The ecosystems of such lakes are entirely isolated from atmospheric influence, including sunlight. However, they are still far from sterile, and additional geothermal activity provides a source of energy for some of these lakes [85]. 

There are multiple logistical challenges for direct sampling of subglacial lake water and sediment. In a study of Lake Whillans, it was revealed that the majority of bacteria present within the water column belong to *Betaproteobacteria*, as well as to *Actinobacteria* and *Firmicutes,* with the dominant annotated OTUs being affiliated with *Polaromonas glacialis*, *Sideroxydans lithotrophicus*, *Albidiferax ferrireducens*, and *Candidatus Nitrotoga arctica*, as well as to the related taxa *Methylobacter* and *Thiobacillus* [86]. The phylum *Thaumarchaeota* dominated the archaea, with the most commonly annotated OTUs being tightly affiliated with *Candidatus Nitrosoarchaeum* and *Nitrosospira multiformis*. Along with the known pool of nitrogen compounds in Lake Whillans, this points to nitrification as an active process. Researchers conclude that the *Betaproteobacteria Candidatus Nitrotoga* plays a key role in the nitrogen cycle, while species of *Sideroxydans*, *Ferriphaselus*, and *Albidiferax* could be implicated in iron cycling.

#### 2.4.2. Meltwater Pond Communities

Meltwater ponds are typically transient and undergoing seasonal freeze–thaw cycles. This means that the properties of these environments depend strongly on the time of year, with the biological activity being highest in the spring and summer when the temperatures are higher, where there is no surface ice, and the winds contribute considerably to the mixing [87]. During winter, as temperatures decrease, a general shift within the community is observed from autotrophy to heterotrophy, with an overall decrease in diversity [88].

Stratification of the physicochemical properties and the communities is also a characteristic of meltwater ponds. For example, *Bacteroidetes* have been seen dominating the surface waters of ponds in Bratina Island, followed by *Betaproteobacteria*, while *Gammaproteobacteria* was the overall dominant taxon at greater depths [87]. A single annotated OTU accounted for more than half of *Bacteroidetes* sequences, namely the marine bacterium *Algoriphagus yeomjeoni*, while most *Proteobacteria* were represented by *Hydrogenophaga taeniospiralis*. *Psychromonas* sp. was the most strongly affiliated taxon in the deeper parts of the pond. A similar study implicated the dominance of *Betaproteobacteria* with *Flavobacterium segetis*, the genera *Bordetella*, *Algoriphagus*, and *Varivorax*, and some other genera from other phyla, such as *Pedobacter*, *Ornithinimicrobium*, and *Loktanella* [88]. Diversity at the surface was also far lower than that of the deeper samples. The dominance of such a small group of species on the surface could be explained by the strong selective pressure of the freeze–thaw cycles that meltwater ponds go through, ultimately selecting bacteria capable of surviving and rapidly re-establishing themselves under these conditions [82]. This is further supported by a study comparing pond samples taken within a one-year gap over time and showing similar dominant OTUs with comparable abundances, which points to a consistent microbial assemblage capable of surviving throughout the year [88]. Overall, the most significant variation in pond community composition throughout the year was in the abundance of a small set of cosmopolitan OTUs, pointing to eolian transport as an essential inoculation factor for all the sampled ponds.

Conductivity and pH are the main physicochemical factors that drive community composition in meltwater ponds [87,88]. More specifically, high conductivity levels correlate with decreased diversity, as more halotolerant species, such as *Algoriphagus* sp., *Ornithinimicrobium pekingense*, and *Loktanella* sp., overtake the community. However, salinity and pH in meltwater ponds experience a gradient throughout the water column and also vary depending on the season, which usually leads to a more complex correlation with community composition. For example, salinity values increase in depth, leading to a higher abundance of the less salt-tolerant bacteria, such as *Hydrogenophaga* sp. and *Algoriphagus yeomjeoni*, on the surface. Both salinity and pH show a decrease in summer samples, which is most likely due to the ponds having been thawed for only a short period of time, meaning less photosynthesis had taken place at the time of sampling. Physiochemical factors, such as pH, NO_2_, Hg, Fe, and PO_4_, also have an influence, but it was not as pronounced as that of conductivity and appeared to be primarily pond specific. However, even with varying abundance, the overall phylum structure was observed to be similar between ponds of different conductivity.

#### 2.4.3. Subglacial Water Communities

Subglacial groundwater systems have the potential to transport a significant number of dissolved compounds and microorganisms while also contributing to continental water discharge budgets [89,90]. This water accumulates at the glacial bed due to surface flow, friction, pressure melting, and groundwater infiltration, but it can flow out to the surface as well. In one such case, hypersaline subglacial brine flowed to the surface of the Blood Falls, Taylor Glacier, which was dominated by *Proteobacteria*, *Bacteroidetes*, *Actinobacteria*, and *Atribacteria*, and less by *Tenericutes*, *Firmicutes*, and *Parcubacteria*. *Gammaproteobacteria* were very abundant, with a dominant annotated OTU affiliated with *Rhodanobacter*. Other prominent OTUs were most closely aligned with *Thiomicrorhabdus* spp., *Desulfocapsa thiozymogenes*, and *Geopsychrobacter electrodiphilus*. *Bacteroidetes* content was represented mainly by clone BF99_C27, *Lutimonas vermicola*, *Aestuariicola saemankumensis*, *Lutibacter maritimus* S7-2. Among archaea, the phylum *Pacearchaeota* was identified. Interestingly, with samples taken a decade apart, the community composition was very similar, showing these subglacial aquafers to be strongly resistant to changes in their environment.

### 2.5. Marine Communities

Bacteria and archaea dominate the oceans’ biomass and play a crucial role in both the production and the degradation of organic compounds [66]. Despite the apparent challenge of low temperatures, access to sunlight, and organic matter, the microbial diversity and richness of the Southern Ocean are comparable to those of other oceans [91]. 

A survey of the marine microorganisms around the west Antarctic Peninsula, Bransfield Strait, covering a large sampling transect of about 500 km at varying depths, strongly correlated depth with community composition, richness, and diversity [92]. Deep Antarctic marine waters (at about 100 m) have higher bacterial and archaea richness and diversity than surface water [92,93]. The same trend is observed when comparing winter and summer samples, the former having much higher diversity and richness [91]. 

The seasonal disparity between surface communities during the winter and summer results from shifting water masses [93]. During summer, meltwater from sea ice and glaciers forces the ‘winter water’ to move down into the water column, stratifying the ocean surface. Due to the lack of competition and the increased sunlight, this new, exposed layer of water is a suitable environment for the development of heterotrophic surface communities, which rely on the degradation of organic compounds, produced by photoautotrophs. Similarly, at increased depths, where access to sunlight and organic compounds is restricted, and there is a seasonal homogeneity in water masses, the communities are dominated by potentially chemolithoautotrophic bacteria and archaea. Surface communities from different locations in both the North Pole and the South Pole have been documented to differ more than those of the deep water [94]. This is presumed due to environmental drivers carrying less weight in the absence of light and in more dormant conditions.

Luria et al. noted that the distance between sampling locations did not strongly impact the community composition of surface waters on the western coast of the west Antarctic Peninsula [93]. However, the study’s authors also proposed that this horizontal homogeneity could be disturbed by the shifting of environmental factors that follow the seasons’ succession. Signori et al. supported this hypothesis, as their study of samples of a nearby area during the transition to fall revealed varying surface bacterial communities at different spatial intervals [92].

Shoreline proximity has a notable impact on community composition. For example, the genera *Roseobacter* and *Zavarzinella* and members of the phylum *Planctomycetes* are more prevalent near the shore [93]. Interestingly, coastal communities are more similar between the North Pole and the South Pole than the deep ocean communities, especially in winter [94]. Limnic phylotypes, such as *Betaproteobacteria*, *Rhizobacterales*, and *Actinomycetale*, in the waters around the Kerguelen Islands suggest a terrestrial influence on bacterioplankton community composition [91].

Surface waters above 100 m are dominated by *Gammaproteobacteria*, *Bacteroidetes*, and *Alphaproteobacteria* [92,93,95]. This profile of the dominant phyla is similar to that of more temperate oceans [91]. Marine waters below 100 m have a significant proportion of archaea (up to 50% of the annotated OTUs), but during winter, they are also present at the ocean surface, most likely due to the lack of stratification [92,93]. They are represented mainly by the genus *Nitrosopumilus*, belonging to the phylum *Thaumarchaeota*, but also *Thermoplasmata*, Marine Groups II and III of the phylum *Euryarchaeota*. *Thaumarchaeota* has been pointed to have a role in nitrification, while *Euryarchaeota* could play a role in protein and lipid degradation [92,96]. This heterogeneity of archaea along the water column could be partially attributed to photoinhibition, which has been documented for *Thaumarchaeota* [97]. In a separate survey, Groups II and III of *Euryarchaeota* are also found in the Amundsen and Ross Seas [98]. The latter study highlights the high archaeal diversity within Antarctic circumpolar deep waters, which also had the highest temperature and salinity of the sampled regions.

The dominance of *Gammaproteobacteria*, especially the orders *Oceanospirillales* and *Alteromonadales*, could be due to their known role in degrading organic carbon through extracellular hydrolytic enzymes [99] or their potential role as chemoautotrophs [100]. Similarly, the large number of *Bacteroidetes* at the surface is indicative of the known role of some species in degrading high molecular weight compounds [66]. However, members of both *Gammaproteobacteria* and *Alphaproteobacteria* can degrade monomers [65]. In addition, specific communities of *Gammaproteobacteria*, *Alphaproteobacteria*, and *Bacteroidetes* have been implicated in sequentially degrading algal biomass [101]. 

In marine ecosystems, *Polaribacter* is a prevailing genus of *Bacteroidetes* [66,91,92,102,103], and its abundance in surface summer waters is attributed to the melting of sea ice, which leads to phytoplankton blooms [102]. In addition, its abundance is also linked to its ability to degrade ocean polymers. Like other *Bacteroidetes*, members of the genus also possess genes for adhesion and light sensitivity [104], which could help their role as heterotrophs in the ecosystems of the Antarctic oceans.

Marine *Alphaproteobacteria* are mainly represented by the order *Rhodobacterales* [92,93,103] but also the order *Pelagibacterales* (formerly SARS11), which has been categorized as a widely distributed marine bacteria [105]. More specifically, the genera *Loktanella* and *Sulfitobacter* have been found in considerable abundance in areas around King George Island [103]. *Deltaproteobacteria* (primarily the SAR324 clade) and *Planctomycetes* were also found almost exclusively in deep waters [92].

Many environmental factors shape the Antarctic marine community composition, such as salinity, oxygen, nitrate, silicate, and phosphate concentrations [92,93,103]. These factors are affected by the vertical stratification in the water column and contribute to the significant impact of depth on community composition.

In a study comparing ocean microbial communities between the North Pole and South Pole, 85% of the annotated OTUs were pole specific. The authors implicated deep ocean currents as a driving force for bacterial similarity [94]. The other 15% were assigned to the clades *Gammaproteobacteria*, *Alphaproteobacteria*, and *Flavobacteria*, some of which could be assigned to the genera *Polaribacter* and *Octadecabacter*. At the same time, deep water communities were dominated by *Deltaproteobacteria*, a typical deep ocean phylotype.

### 2.6. Microbial Mat and Sediment Communities

The combination of organic and inorganic sediments at the bottom of water bodies can be a suitable interface for the development of microbial mats, which contain bacteria and algae in a typical layered structure, with photosynthesizing aerobes on the surface and heterotrophic organisms deeper in the mat. However, mats can also appear on the soil/air interface if conditions, mainly humidity, are suitable. Sediment samples on their own, without the presence of a mat, can harbor a different microbial assemblage than that of the open water.

#### 2.6.1. Community Composition

The dominant phyla within the microbial mats are typically *Cyanobacteria*, focused on the surface of the mat, with a more diverse set of bacteria, algae, and even microinvertebrates in the center [67,72,106]. Thus, *Cyanobacteria* provide substrates and nutrients for the rest of the community, which in turn are capable of degrading those compounds. However, multiple studies of the Antarctic mats show that *Proteobacteria* (mostly *Alphaproteobacteria* and *Betaproteobacteria*) and *Bacteroidetes* are often equally found, if not more abundant [7,107,108,109,110]. In fact, a study comparing the Artic and Antarctic bacterial mats revealed the same high abundance of *Proteobacteria* on both poles and similar overall community composition [110]. 

*Bacteroidetes* can reach high abundance in mats worldwide, including Antarctica, where the phylum is most common in stream meltwater [108,109]. *Bacteroidetes* species are capable of degrading a large array of organic matter and can have species-specific interactions with *Cyanobacteria* and diatoms in mats [111,112].

*Phormidium autumnale* and *Leptolyngbya antarctica* have been identified as the dominant filamentous *Cyanobacteria* in the top layers of the Antarctic mats [106,108,113,114,115] where they function as photoautotrophs, providing carbon for the lower layers of the mat, inside which a more diverse set of heterotrophic bacteria partake in the carbon cycle. Members of the order *Pseudanabaenales* can also be identified at higher abundances (greater than 50%), especially when *Cyanobacteria*-specific primers are used in metagenomic studies [115]. Despite their importance in nitrogen cycling, in the molecular sequencing surveys of microbial mats, members of the order *Nostocales* are relatively underrepresented, rarely exceeding 10% of the OTUs [115]. 

*Firmicutes*, *Actinobacteria*, *Chloroflexi*, *Verrucomicrobia*, and *Thermus*–*Deinococcus* are also frequent within the Antarctic mats but rarely exceed 10% of the OTUs. The high levels of *Actinobacteria* in some mats, like those of Lake Untersee, Northern Antarctica, have been attributed to their role in decomposing cyanobacteria from the top layers of the mats [114]. The same lake contained a great abundance of *Verrucomicrobia*, especially *Opitutus* spp., including *Opitutus terrae*, members of which have been reported in similar cold anaerobic environments with a role in nitrogen fixation and as producers of acetate and propionate, contributing to methane cycling [116,117,118]. This could point to their role within the inner layers of the benthic microbial mats, where along with the depth of the mat, they are influenced by the oxygen gradient.

The results from a metagenomic study of ancient paleomats embedded inside the valley walls of the McMurdo Dry Valleys could elucidate the shifts in the mat community structure that took place on the scale of thousands of years [119]. The findings suggest that, compared with modern mats, the paleomat contains indigenous mat cells capable of flourishing under specific conditions. The high amount of heterotroph-annotated OTUs, which are also found in surrounding soils, suggests the use of the deposited paelomat layers as a carbon source for colonizing soil bacteria. More specifically, *Actinobacteria* (order *Actinomycetales*), *Bacteroidetes* (orders *Cytophagia* and *Sphingobacteriales*), *Firmicutes* and *Proteobacteria* (*Alpha*- and *Gammaproteobacteria*), *Gammatimonadetes*, *Chloroflexi*, and *Deinococcus*-*Thermus* dominated the paleomats. The modern mat was composed primarily of *Proteobacteria* (*Beta*-, *Alpha*-, and *Gammaproteobacteria*), followed by *Bacteroidetes* and *Cyanobacteria*, while *Actinobacteria* of nearby soil comprised approximately 60% of the reads. The results show that heterotrophic, autotrophic, and nitrogen-fixing bacteria are metabolically active in paleomats. The same study identified genes for stress response and DNA repair but also antibiotic resistance within the paleomat *Proteobacteria* group, hinting at competition as a considerable driver of community composition, even in the challenging conditions of paleomats.

A comparative study of the mats in different ponds in the McMurdo Dry Valleys based on both 16S and 18S sequencing found that eukaryotes had lower community richness and less diversity between sample sites compared to bacteria and archaea [107]. There was also a strong correlation between bacteria and archaea communities, pointing to either species-specific interactions within these groups or habitat filtering leading to high homogeneity across the ponds. The same survey also found a high abundance of the *Cyanobacteria* family *Nostocaceae*, implicating their role in the mats with nitrogen fixation. The dominant *Nostocaceae* genera shifted from *Nostoc* to *Nostocales* along with the increasing salinity. Interestingly, *Oscillatoriales* were also present at all ponds but with different dominant species, depending on the site, suggesting that this group could function as a reservoir of diversity capable of persisting in varying conditions and aiding the mat’s resistance to changes in environmental factors.

Glacial meltwater streams also possess well-developed mat communities. A study showed them to be dominated by *Cyanobacteria*, *Bacteroidetes*, *Proteobacteria*, *Acidobacteria*, and *Firmicutes*, with *Proteobacteria* being the most diverse group [120]. *Planctomycetes*, *Verrucomicrobia*, *Thermi*, *Armatimonadetes*, and *Actinobacteria* were less abundant. Filtering out *Cyanobacteria*, the most abundant bacterial orders were *Sphingobacteriales*, *Burkholderiales*, and *Chloroacidobacteria*. The dominant *Cyanobacteria* taxa depended on the mat types and stream location: black mats—*Nostoc* or *Pseudanabaenaceae*; green mats—*Synechococcophycideae*; orange—*Microcoleus*; red—*Leptolyngbya*. The same study also found *Ascomycota* and *Basidiomycota* as the dominant fungal phyla. Meltwater stream mats also boast very high diversity and are generally recognized as diversity hotspots. This is best explained by the fact that meltwater streams are highly heterogenous environments and contain regions of varying substrate availability, stream flow rate, and other conditions, which give rise to a high number of environmental niches, thus promoting diversity. The impact of chemical factors, such as the pH and conductivity, was marginal, which could be explained by the less oligotrophic condition of meltwater streams compared to stationary bodies of water, which are more susceptible to environmental influences. Additionally, some of the dominant phyla, such as *Acidobacter*, a common bacteria in soils of the McMurdo Dry Valleys, show that meltwater streams are at least partially seeded by surrounding soils [54].

One metagenomics study focused on the fungi from lake sediments from Vega Island and found high richness and diversity, with the dominant phyla being *Ascomycota*, *Rozellomycota*, *Basidiomycota*, *Chytridiomycota*, and *Mortierellomycota*, and the most abundant genera being *Pseudogymnoascus*, *Penicillium,* and *Mortierella*. Psychrotolerant, cosmopolitan decomposers and pathogens made up most of the annotated amplicon sequence variants [121]. 

Sediments of the subglacial Lake Whillans revealed *Betaproteobacteria* to be the dominant class, gradually being replaced by *Gammaproteobacteria* with the increase in sediment depth [86]. The most common sediment annotated OTUs were affiliated with *Smithella propionica*, *Ignavibacterium album*, and Candidate division JS1. *Euryarchaeota* was the major archaeal phylum of the sediments with the most abundant OTU, closely matching *Methanohalophilus levihalophilus* and *Methylobacter tundripaludum*. Other dominant OTUs in the sediment were affiliated with *Sideroxydans lithotrophicus* and *Albidiferax ferrireducens*. The annotated OTUs present within the lake generally had a very low abundance in the sediment. The bacteria implicated in sulfur cycling, such as *Desulfatiglans*, *Desulfatibacillum*, *Sideroxydans*, and *Thiobacillus,* accounted for a considerable number of sequences in the study.

Deep-sea sediments are dominated by a diverse heterotrophic bacterial assemblage, primarily represented by the phylum *Proteobacteria*, most commonly *Alphaproteobacteria* from *Rhodospirillaceae* (especially the genus *Pelagibius*) and *Rhodobacteraceae* (genera *Loktanella* and *Litorimicrobium*) and *Gammaproteobacterial* NOR5/OM60 (*Haliea*/*Congregibacter*), *Sinobacteraceae* (‘Marine Benthic Group’), *Moraxellaceae* (mostly genus *Psychrobacter*), *Piscirickettsiaceae* [122,123,124]. Far less abundant are the phyla *Firmicutes*, *Bacteroidetes*, *Actinobacteria*, *Chloroflexi*, *Planctomycetes*, *Verrucomicrobia*, and *Acidobacteria*. Archaea in deep-sea sediments are represented by *Nitrosopumulis*-type *Thaumarchaeota*, the abundance of which correlated negatively with organics and nutrient content in the water but was overall high for archaea. *Flavobacteria* were also detected in considerable abundance, mainly due to a single annotated OTU affiliated with *Psychroserpens*. In these waters, researchers highlight the critical role of the phylum NOR5/OM60 in the degradation of algal detritus. In methane-rich zones, the phylum *Atribacteria* dominates, and its abundance correlates with the amount of methane and sediment depth, and the phylum most likely plays a role in the methane cycle in these anoxic conditions [125]. In the same methane-rich areas, the rest of the community included *Bacteroidetes*, *Proteobacteria* (mainly *Alpha*- and *Gammaproteobacteria*), *Planctomycetes*, *Actinobacteria*.

Fungi in deep-sea sediments are quite diverse and dominated by the phyla *Ascomycota*, *Basidiomycota*, *Mortierellomycota*, but also including *Mucoromycota*, *Chytridiomycota*, and *Rozellomycota*, with the most prominent genera being *Mortierella*, *Penicillium*, *Cladosporium*, *Pseudogymnoascus*, *Phaeosphaeria,* and *Torula* [121]. The study showed a complex system of fungi inside deep Southern Ocean sediments with high metabolic diversity, including symbionts, decomposers, and pathogens. 

The extremely hot Deception Island soil mats (88 °C) contain only one identifiable archaeon, belonging to the phylum *Thaumarchaeota* and comprising about 35% of sequences. Representatives of the same order were found with a similar abundance in cooler mats in the area (8 °C) but were barely present in the coldest mats (2 °C). The latter was dominated almost entirely by the phylum *Euryarchaeota*, which represented 96% of sequences. Order *Nitrosopumilales* were also detected in the two cooler samples but at very low abundance [53].

#### 2.6.2. Factors Impacting the Mat Communities

The hydrologic regime (the flow intensity and flow consistency of meltwater streams) impacts the co-occurrence of heterotrophic bacteria and diatoms within the mats but not that of *Cyanobacteria* and diatoms [108]. This is most likely due to the intense selective pressures on the surface of microbial mats, where fluctuating light, temperature, and moisture select highly specialized *Cyanobacteria*. However, in the same study, *Cyanobacteria* and heterotrophic bacterial diversity had an overall inverse correlation with diatom diversity. 

In a global study of *Cyanobacteria*, where the McMurdo Dry Valleys in Antarctica were sampled, the location was the best predictor of *Cyanobacteria* diversity within the valley. Still, temperature also had a positive correlation [113]. The red and red/green samples were dominated by the family *Pseudanabaenaceae*, while the black samples by *Phormidiaceae*. *Phormidium murrayi* and *Leptolyngbya frigida* found in the study are typical species for other cold regions, reinforcing the hypothesis that cold tolerance at different locations evolves at a similar speed [126].

In a study comparing McMurdo Dry Valleys’ microbial mat communities exposed to different environmental factors, salinity was the best predictor for community composition, followed by temperature and pH [107]. More specifically, *Cyanobacteria* dominated the mats in high salinity ponds, with Pseudanabaenales replacing *Phormidiaceae* as the dominant family in these conditions. *Proteobacteria* were also highly abundant and more diverse at higher salinities than *Cyanobacteria*, with *Alteromonadaceae* and *Rhodobacteriaceae* replacing *Comamonadaceae* as the dominant family.

In a much broader geographical study of the lacustrine mat communities in ice-free Antarctic regions, surveying the Antarctic Peninsula, McMurdo Dry Valleys, and east Antarctica using *Cyanobacteria*-specific primers, it was found that community composition was dependent primarily on salinity and did not correlate strongly with location [115]. Additionally, half of the identified *Cyanobacteria* sequences were considered potentially endemic, while other cosmopolitan species that were widespread throughout the continent represented a much smaller portion of sequences. These results point to a more complex dispersal of the organisms, where the effect of the geographical barriers has a small contribution compared to the habitat conditions.

Microbial mats have a considerable dispersive potential. Even in permanently ice-covered lakes, microbial mats are capable of breaking off from the benthic surface in ‘lift-off’ events due to the high O2 concentration present [127]. Once they reach the ice sheath that covers the surface, they are incorporated and gradually, over the years, migrate upward through the ice column to eventually reach the surface and be dispersed by winds. This is possible due to the high tolerance of microbial mats toward desiccation, which was shown by the rapid reactivation of a dry microbial mat in the McMurdo Dry Valleys when exposed to water [128].

### 2.7. Cryoconite Hole, Glacial Ice, and Snow Communities

The meltwater of lake ice cores contains metabolically active bacteria that flourish during optimal growth conditions, the temperature playing the most important role [82]. However, glacial ice is a highly oligotrophic environment, as the low diffusion of nutrients and low temperatures severely hamper the growth of all microorganisms [129].

#### 2.7.1. Cryoconite Hole Communities

Cryoconite holes are formed on glaciers or ice-covered lakes when the dark substrate on the surface, most commonly soil, absorbs heat from the sun fast enough to melt the ice layer underneath [83]. This creates a shallow hole in the ice, containing water plus organic and inorganic sediment at the bottom. Antarctic cryoconite holes are often isolated from the atmosphere through a thin ice lid at the top, and carbon dating shows this isolation to be greater than that of soils [130]. These unique microenvironments, exposed to low light intensity and oxygen, harbor much higher biodiversity than regular glacial ice, even being recognized as microbial hotspots [131]. On the other hand, glacial ice microorganisms reside on the surface of dust particles and liquid water veins inside the ice [132].

Melting ice surrounding cryoconite holes in the Untersee Oasis, east Antarctica, contains *Proteobacteria*, *Cyanobacteria*, *Bacteroidetes*, *Actinobacteria*, *Acidobacteria*, *Verrucomicrobia*, and *Gemmatimonadetes* in a reasonably even abundance [83]. The lake bare-ice samples from the same location contained a smaller diversity of phyla, with *Proteobacteria* representing about 60% of the annotated OTUs, followed by *Actinobacteria*, *Bacteroidetes*, and *Cyanobacteria*. On the other hand, *Cyanobacteria* dominated white ice patches, followed by *Proteobacteria* and *Bacteroidetes*. The genera *Tychonema*, *Leptolyngbya*, and *Chamaesiphon* dominated the *Cyanobacteria* in both ice types. Glacial ice samples from the Anuchin Glacier of the oasis contained *Proteobacteria*, *Actinobacteria*, *Firmicutes*, *Bacteroidetes*, and *Cyanobacteria*, the last of which shared genera with the lake ice samples. The dominant *Cyanobacterial* genera in glacial ice were *Chroococcidiopsis*, *Geitlerinema*, *Microcoleus*, *Nostoc*, *Phormidium*, and *Pseudanabaena*. However, there were differences in *Cyanobacteria* genera between the sampling locations, indicating that local conditions or biotic sources may influence community composition. Interestingly, in this particular study of Lake Untersee, cryoconite holes had less microbial diversity than the surrounding glacial ice, which could be attributed to the isolated nature of lidded cryoconite holes. At the same time, community composition was significantly different between cryoconite holes and the surrounding glacial ice.

Another study of cryoconite holes and the surrounding area in Queen Maud Land again revealed *Proteobacteria*, *Cyanobacteria*, and *Actinobacteria* as the dominant phyla, followed by *Bacteroidetes* and *Acidobacteria* [130]. *Cyanobacteria* were represented mainly by *Oscillatoriophycideae*, more specifically, *Xenococcaceae* and *Phormidiaceae*. The majority of sequences could not be assigned to lower classifications. A single archaeon, *Nitrososphaera*, was detected. It was also revealed that community composition was more similar within a cryoconite hole than with the other sample types. In the same study, community composition was best explained by the amounts of NO_3_^−^, SO_4_^2−^, Cl^−^, and total solid carbon, while relative abundance depended heavily on location, suggesting local selective pressures. 

Interestingly, the community composition of cryoconite holes shares more common genera with the surrounding glacier ice than with the air and soil, despite cryoconite holes being inoculated primarily by soil particles and often ending up with air trapped under an ice lid [83]. However, some studies show that biological inoculation of the holes depended on the sample site, and it was attributed to soil, snow, and lake water in different ratios, with the exception that sampled endoliths had a minor contribution to all cryoconite holes [130].

#### 2.7.2. Ice and Glacial Microbial Communities

Sequencing of the accretion ice of Lake Vostok, formed by the freezing of the lake water, revealed the dominant bacterial phyla as *Firmicutes*, *Proteobacteria* (mainly *Gamma*-, *Beta*-, and *Alphaproteobacteria*), *Bacteroidetes*, and *Actinobacteria*, while fungi were represented by *Ascomycota* and *Basidiomycota*. Only two methanotrophic archaea species were detected, both also belonging to the deep ocean sediments and adapted to high-salt marine environments—*Halorubrum trapanicum* and *H. salinarum* [85]. Many psychrophile *Gammaproteobacteria* sequences were identified but also psychrophilic and psychrotolerant sequences of *Actinobacteria*, *Alphaproteobacteria*, *Bacteroidetes*, *Betaproteobacteria*, *Firmicutes*, and archaea. Sequences close to those of sulfur oxidizers, alkali-tolerant bacteria, *Jeotgalicoccus halotolerans*, *Nesterenkonia halotolerans*, and other halophilic bacteria were also identified within the ice. Interestingly, marine species were also found, such as *Pseudomonas xanthomarina.* The co-occurrence of marine, halophilic, and halotolerant species suggests highly saline layers of water present within the lake, which could point to Lake Vostok having a direct connection with the sea roughly 35 million years ago when sea levels were much higher, and the region was ice free. The hydrothermal activity explains the inclusion of thermophiles near Lake Vostok’s shallow embayment. Nitrogen-fixing bacterial species belonging to *Azospirillum*, *Azotobacter*, *Bacillus*, *Burkholderia*, *Cyanobacteria*, *Frankia*, *Klebsiella*, *Rhizobium*, *Rhodobacter*, *Rhodopseudomonas*, and *Sinorhizobium*, along with nitrifying bacteria from the genera *Methylococcus*, *Nitrobacter*, *Nitrococcus*, *Notrosococcus*, and *Nitrosomonas*, were also identified. A total of 95% of the species richness was attributed to decomposers, autotrophs (both nitrogen and carbon fixing), and nitrogen cycling bacteria. 

All steps of nitrogen cycling in Lake Vostok’s basal and accretion ice were represented by at least one species [85]. Secondary functions, such as iron oxidation, iron reduction, arsenite oxidation, and arsenate reduction, were also present. *Actinobacteria*, such as *Mycobacterium* and *Streptomyces*, were implicated in nitrogen fixation, nitrification, and denitrification, while decomposition was attributed to *Ascomycetes* and *Basidiomycetes*. *Proteobacteria* of the genera *Azoarcus*, *Nitrotoga*, *Nitrosococcus*, *Nitrosomonas*, and *Pseudomonas*, as well as *Clostridales* from *Firmicutes*, also most likely contribute to nitrogen cycling. Many of the ice autotrophs were chemolithoautotrophs, utilizing hydrogen, sulfur, nitrogen, or iron as an energy source.

While *Cyanobacteria* represented most of the large phototroph diversity inside Lake Vostok accretion ice [133], they cannot rely on photosynthesis for primary production and most likely survive as heterotrophs [134]. Furthermore, a similar study, sampling the upper layers of basal and glacier ice on top of the accretion ice of Lake Vostok, found that the sequences were dominated by uncultured *Cyanobacteria* [85]. The same *Cyanobacteria* were also found in the lower layers of the ice but at a considerably reduced abundance, and contamination could not be ruled out in the study. 

A comparison between basal ice (formed from the accumulation of snow) and accretion ice above Lake Vostok revealed the two samples have very few organisms in common, highlighting the extraordinarily slow migration of microorganisms through the ice column [85]. However, the sudden and unfavorable shift in environmental conditions between the basal and accretion ice gradient also contributes to this discrepancy. 

Bacteria inside glaciers are represented by the phyla Proteobacteria (mainly Gammaproteobacteria and Alphaproteobacteria), Bacteroidetes (class Bacteroidia, order Flavobacteriales), Firmicutes, Actinobacteria, Acidobacteria, Verrucomicrobia, Planctomyetes, and Gemmatimonadetes with prevalent genera Flavobacterium, Luteolibacter, Rhodoferax, Rhodanobacter, Dokdonella, and Polaromonas [52,135]. Typical Antarctic psychrotrophic and psychrotolerant bacteria in glaciers include the genera Polaribacter, Polaromonas, Psychrobacter, Psychromonas, and Cryobacterium. Thermoplasmata from phylum Euryarchaeota was the dominant archaeal class, represented only by Marine Group II and Methanomassiliicoccus as the closest sequenced relative. The ammonia and the total nitrogen had the highest impact on community composition, with high levels associated with Nitrospirae. Overall, glaciers contained distinct psychrophilic, methylotrophic, denitrifying, and nitrifying groups. Although the stratification of glacial communities has not been exclusively studied, deep glacial samples (3 m) seem to exhibit a different microbial assemblage with Actinobacteria, Bacteroidetes, Cyanobacteria, Proteobacteria, and Firmicutes in order of abundance [136].

Recently, a study of glacial ice in the Antarctic Peninsula and the South Shetland Islands focused on fungi revealed *Ascomycota* as dominant, followed by *Basidiomycota* and *Mortierellomycota* [129]. The dominant genera and species (greater than 10% of reads) were *Penicillium* sp., *Cladosporium* sp., *Penicillium atrovenetum*, *Epicoccum nigrum*, *Pseudogymnoascus* spp., *Phaeosphaeriaceae* sp., and *Xylaria grammica*. Interestingly, the diversity index of the fungi was high, with a small number of sequences being present in all samples. Most of the sequences were previously unreported in Antarctica or could not be classified to narrower taxonomic levels, pointing to glacial ice as a possible underexamined source of fungal diversity. Unsurprisingly for glacial conditions, most of the fungi were saprophytic. Many of the dominant taxa were previously detected within Antarctic aerosols and most likely precipitate with snowfall and over time, being compacted into the glacial ice. This high fungal diversity index and similar community composition have been observed for snow near Livingston Island, using the same ITS3 and ITS4 amplicon sequencing approach [137]. The dominant phyla were again *Ascomycota*, *Basidiomycota*, and *Mortierellomycota,* with most of the taxa abundance being attributed to *Cladosporium*, *Pseudogymnoascus*, *Penicillium*, *Meyerozyma*, *Lecidea*, *Malassezia*, *Hanseniaspora*, *Austroplaca*, *Mortierella*, *Rhodotorula*, *Penicillium*, *Thelebolus*, *Aspergillus*, *Poaceicola*, *Glarea*, and *Lecanora*, some of which are known cold-tolerant, cosmopolitan, and opportunistic taxa. A large proportion of fungi in both studies remains identifiable only to higher classifications, which supports the idea that much of the fungal diversity of Antarctica remains poorly understood.

Fungi inside the permafrost have also been surveyed using NGS-based metagenomics with annotated phyla, the order of abundance being *Ascomycota*, *Mortierellomycota*, *Basidiomycota*, *Chytridiomycota*, *Rozellomycota*, *Mucoromycota*, *Calcarisporiellomycota,* and *Zoopagomycota* [138]. The dominant species and genera were *Pseudogymnoascus appendiculatus*, *Penicillium* sp., *Pseudogymnoascus roseus*, *Penicillium herquei*, *Curvularia lunata*, *Leotiomycetes* sp., and *Mortierella* spp. The large diversity of fungi even included human pathogens, and the researchers outlined the risk of their potential release due to melting from increasing average global temperatures. Additionally, out of the physicochemical factors, pH, K, Al, P had the most significant impact on community composition.

#### 2.7.3. Snow Microbial Communities

*Proteobacteria*, and more specifically, *Alphaproteobacteria*, are by far the most commonly identified taxon in Antarctic surface snow, followed by *Bacteroidetes* [139,140,141]. *Cyanobacteria*, *Firmicutes,* and *Actinobacteria* have also been found in lower abundance. Studies of the surface snow near the Russian research stations revealed no consistent bacterial community, with the overall dominant classes being *Alphaproteobacteria*, *Betaproteobacteria*, *Gammaproteobacteria*, *Sphingobacteriia*, *Flavobacteriia*, *Cytophagia*, *Actinobacteria*, *Chloroplast*/*Cyanobacteria*, and *Bacilli* [139]. Still, the genus *Flavobacteria* was most abundant in all samples, followed by the genera *Janthinobacterium*, *Ralstonia*, *Hydrogenophaga*, *Pseudomonas*, *Caulobacter*, *Acinetobacter*, and *Comamonas*. A similar study also found *Proteobacteria* and, more specifically, *Alphaproteobacteria,* to dominate the surface snow, followed by *Bacteroidetes* and *Cyanobacteria* [141]. Although only half of *Alphaproteobacteria* sequences have been identified, the dominant taxa were *Kiloniellaceae*, *Rhodobacteraceae*, and *Rhodospirillaceae*, which points to a strong marine influence on the snow community. A small number of sulfur reducers, such as *Desulfobacterales*, were identified among *Deltaproteobacteria*, while *Bacteroidetes* were primarily represented by *Flavobacteriaceae*. Researchers noted that the community composition was similar to ones reported from glaciers, snow, lake ice, sea ice, and atmospheric clouds.

The spatial distribution of Antarctic snow microbiotas has also been noted. In a large-scale study of surface snow along the Antarctic Peninsula, starting from the South Orkney Islands to the Ellsworth Mountains, a higher abundance of *Firmicutes* was found near continental Antarctica. In contrast, more *Actinobacteria* and *Bacteroidetes* were found in Antarctic island snow [140]. Similarly, *Bacteroidetes* shifted from *Cytophaga*, *Flavobacteria*, *Sphingobacteriia*, and *Chitinophagia* at northern sites to only *Chitinophaga* in the south but with a high abundance of unidentified sequences at Sky Blu and Pine Island Bay. On the genus level, continental Antarctica represented considerably more *Methylobacterium* and *Rhizobium*. These differences result from the spatial distance, the seasonality of the snow cover, snow properties, and the overall difference in climate between the sampling sites. The majority of organisms were also identified as soil related, but this was likely not the result of contamination. It was also shown that while soils were the primary biological inoculant for surface snow, the impact of aerosol microorganisms is much more significant for continental inland Antarctica. Microbial communities also varied more in continental Antarctica, even with the same location, which could be attributed to the harsher conditions and weather.

### 2.8. Airborne Microorganisms

Although there have not been many NGS-based metagenomics studies of the airborne microorganisms of Antarctica, some trends do emerge from the few studies available. A joint study of soil and air in the McMurdo Dry Valleys revealed that the two environments had a different community composition, which points to the dispersal power of wind, which is negligible, at least in the area of the McMurdo Dry Valleys [142]. Furthermore, the bacteria present in the air did not match those of the surrounding sea and lakes, and the most abundant taxa were common to those of the global aerosols. However, most of the bacteria originated from Antarctica, since the airmasses came predominantly from the Antarctic Plateau. This discrepancy could point to strong selective pressure on the soil. Aerosols were dominated by *Firmicutes* (mostly *Bacillus* and *Clostridia*) and *Proteobacteria* (*Alpha*-, *Beta*-, and *Gammaproteobacteria*), with most of the dominant bacteria being thermophilic. Similar community composition is found near lakes within the bioaerosols, such as that of the Untersee Oasis, more specifically, members of the genera *Staphylococcus*, *Bacillus*, *Corynebacterium*, *Micrococcus*, *Streptococcus*, and *Neisseria* [83].

A large-scale study comprising the entire Antarctic coastal water perimeter determined *Proteobacteria*, predominantly *Alpha*- and *Betaproteobacteria*, as representing about 90% of sequences, and *Firmicutes* with about 1.5% [143]. Almost half of the sequences were determined to have a terrestrial origin, with only 4% having a marine origin, despite the sampling taking place at sea. The air masses came mainly from the continent.

Airborne fungi belonging to the phyla *Ascomycota*, *Basidiomycota*, and *Mortierellomycota* were found using the ITS3 and ITS4 amplicon sequencing on Levingston Island [137]. The dominant taxa were *Pseudogymnoascus*, *Cladosporium*, *Mortierella*, and *Penicillium* sp. When comparing the community composition of the air and snow in the area, it was found that more sequences were shared between the two environments than the sequences unique to each of them. Furthermore, no sequences could be attributed solely to the air samples when considering only the dominant taxa. This shows that snow communities are not merely the result of biological influx from the air but are instead an active ecosystem.

### 2.9. Impact of Macro Flora and Fauna on Microbial Communities

Despite the scarcity, the Antarctic ecosystems contain various animals and even higher plants that have an undeniable impact on the microbial life on the continent. For example, compared to pristine soils, the Antarctic soils covered with vegetation (mosses) possess greater bacterial abundance, total carbon, and total nitrogen [37].

#### 2.9.1. Impact of Higher Plants

Two separate metagenomic studies of soil from the roots of two Antarctic vascular plant species (*Deschampsia antarctica* and *Colobanthus quitensis*) found no notable differences between their rhizosphere bacterial microbiotas [34,144]. In one of the studies, the samples were dominated by *Firmicutes* (mainly *Clostridia*), followed by *Actinobacteria* and *Proteobacteria* (mainly *Gamma*- and *Alphaproteobacteria*), which does not match the typical pattern for temperate climate soils, which are dominated by *Proteobacteria* [145]. However, the second survey showed the dominating phyla to be unclassified, followed by *Bacteroidetes*, *Acidobacteria*, and *Proteobacteria*. However, because both samples were taken from a fairly similar area on King George Island, we believe this apparent mismatch could be attributed to the disparity in sequencing regions, i.e., V4 for the first study and V1–V2 for the latter.

The slight impact of plant species on the bacterial community composition, especially compared to the impact of edaphic factors, could be explained in several ways. As noted in several other studies, the soil may play a more significant role in plant bacterial composition [146,147]. There could also be different environmental factors, such as the climate or soil composition, exerting broader selective pressure on the bacteria compared to that of the plant rhizobiome. Another hypothesis is that the limited species diversity of both plants and bacteria leads to a narrower space for cooperative relationships between them.

Additionally, more unclassified bacteria have been reported within the Antarctic soils with a vegetative cover than within the exposed soils [20,148]. This can be partially explained by the sequences available in the classifier, most of which are derived from temperate climates and which differ from those in extreme environments. Furthermore, the thermal isolation, water retention, and microenvironments created by plant rhizospheres could also reduce the Antarctic climate’s selective pressures and allow for more temperate bacteria to thrive [20].

#### 2.9.2. Impact of Big Animals

When a large animal (such as a seal or penguin) dies, there is typically a high chance its carcass will remain on the spot for a very long time. If no large predator is around to consume it, or if it does not end up in the ocean, the sun, the low moisture, and the low temperature will slow down decomposition and effectively ‘mummify’ the remains. A study focused on the soil beneath a mummified seal carcass has elucidated the complex community interactions, which can develop in those conditions [63]. Indeed, the researchers determined the age of the carcass being about 250 years old and found, when compared with the surrounding soil, that it provided nitrogen and carbon, stabilized the temperature fluctuations, shielded from the UV rays, and raised humidity in the soil underneath, leading to an increase in the direct bacterial cell count, released carbon dioxide, and ATP. Edaphic factors, such as the pH and salinity, were higher in the carcass soil but had a much lower impact on community composition, which was shown by the rapid decrease in cell counts upon moving the body of the dead seal several meters away. Interestingly, the diversity of the bacterial community underneath the carcass was lowered, which was due to two annotated OTUs dominating most of the sequences. On the other hand, most of the control soil bacteria were shared with those of the carcass soil, indicating that the microorganisms native to the seal contributed little to decomposition. These two facts, taken together, could point to the Antarctic soils as a media containing a small number of opportunistic microorganisms that reside in a dormant state. When enough nutrients are provided, such as in the case of a dead animal, those organisms outcompete the other native bacteria. After a five-year period, two dominating annotated OTUs reached up to 80% of the sequences. They belonged to *Psychrobacter* sp. (*Gammaproteobacteria*) and to the family *Planococcaceae* (*Firmicutes*). The same study also showed that bacterial composition within the Antarctic soils can change rapidly over just a few years and that the activity of the native fauna plays a significant role in this process. This influx of *Firmicutes* and *Proteobacteria* in response to the addition of resources into the soil has been confirmed additionally by another study of Antarctic soils [47].

The habitation of penguins in rookeries generates ornithogenic soils, but despite the increased nutrients and the biomass, their biodiversity is comparable to that of nearby mineral soils [35]. *Firmicutes* and/or *Psychrobacter* were found to dominate soils currently occupied by penguins, while *Xanthomonadaceae* and *Actinobacteria* dominated previously occupied soils. *Firmicutes* identified in ornithogenic samples resemble endospore formers, such as *Oceanobacillus profundus* and *Clostridium acidurici*, while *Xanthomonadaceae* was most closely represented by *Rhodanobacter*, *Dokdonella*, and *Lysobacter*. More recently, in an NGS study of active ornithogenic soils from King George Island, *Firmicutes* dominated the community composition with roughly 80% of the sequences, primarily attributed to the genus *Thermohalobacter*, *Tissierella*, *Carnobacteriaceae*, and *Desulfonispora* [64]. However, formerly active soils were dominated by *Bacteroidetes* and *Proteobacteria*. A similar study assessing the vertical community structure of historical ornithogenic sediment cores found that the dominant phyla comprised *Proteobacteria* (most commonly the classes *Beta*- and *Gammaproteobacteria*), *Actinobacteria*, *Bacteroidetes*, *Gemmatimonadetes*, *Acidobacteria*, and *Chloroflexi* but at different relative abundance depending on the depth [149]. The researchers also found that the bacterial richness, diversity, and phyla relative abundances correlated positively with the amount of penguin activity.

#### 2.9.3. Impact of Small Animals

Sea sponges are abundant at the bottom of the Antarctic marine ecosystems, and it is plausible that they have a considerable impact on the benthos community composition. A study of sea-sponge-associated microorganisms of Fildes Bay, King George Island, revealed that the dominant bacteria were *Proteobacteria*, followed by *Bacteroidetes*, *Verrucomicrobia*, *Thaumarchaeota*, and *Planctomycetes*, and all the phyla were detected in the seawater as well [150]. Dominant bacterial orders were *Nitrosomonadales* and *Methylophilales* (mostly sequences from *L. antarctica*). Among the archaea, the Marine Group I sequences were affiliated with *Candidatus Nitrosopumilus*. Overall, bacteria, archaea, and eukaryotes associated with sea sponges displayed higher diversity when compared with seawater assemblages. Rodríguez-Marconi et al. also noted that compared to global trends of sea-sponge-associated microorganisms, the presence of *Chloroflexi* and absence of *Cyanobacteria* and *Poribacteria* were the most striking differences [150].

### 2.10. Endemism

A metastudy of Antarctica soil surveys showed that many *Bacteroidetes* species and some *Actinobacteria* from Princess Elisabeth Station, east Antarctica, could be endemic to Antarctica [45]. A study of the McMurdo Dry Valleys found almost no match between the annotated OTUs detected at the different sites of the valley, which the authors suggest could point to intra-valley endemism within the continent [38]. Endemism has also been implicated in *Cyanobacteria* within microbial mat communities [115]. The varying dispersive power of the Antarctic winds [142] and the lack of other vectors for transportation could mean that specific areas isolated by geography and landscape could harbor the development of endemic species. Indeed, the series of mountains and valleys inside the McMurdo Dry Valleys could give the ideal environment for such a process to occur. Additionally, global warming threatens the diversity of Antarctic endemic species, as they become vulnerable to being outcompeted by cosmopolitan generalists from warmer climates [151]. 

Endemism has also been reported for fungi, and endemic fungal species, such as *Antarctomyces psychrotrophicus* and *Antarctomyces pellizariae,* have been, respectively, isolated from soil and snow on King George Island [152].

In most of the meta-barcoding studies, a large portion of the sequences are not found in the databases used or cannot be classified to a lower taxonomic level. Indeed, even in recent studies, this number is usually very high for fungi, reaching up to 66% of OTUs [121]. This indicates a not yet fully understood assemblage of many Antarctic environments and prompts further investigation of these habitats, since the additional threat of climate change could place this undiscovered biodiversity permanently out of reach. Some grim predictions can also be made because human pathogenic fungi have been found inside the Antarctic permafrost and could pose a potential risk if the increasing global temperatures lead to a mass melting of glaciers and permafrost [138].

### 2.11. Impacts of Climatic Changes and Global Warming on Antarctic Microbial Communities

Global warming undeniably has an enormous impact on the Antarctic ecosystems. It has been well documented that the oceans around the continent are changing rapidly, with rates of warming and freshening exceeding the global ones, leading to a contraction of the typical native ecosystems to higher latitudes [153,154]. These changes also affect deep-sea ecosystems directly (causing shifts in bottom-water temperature, oxygen concentration, and pH) and indirectly, through changes in the surface oceans’ productivity and export of organic matter to the seafloor [155]. The latter authors also state the hypothesis that microbial metabolism can impact the ocean’s ability to buffer the results of global climate changes. 

A good demonstration of how climate changes affect the microbial community is the experimental study of Evans et al. performed on coastal waters [156]. The authors report that microbial communities varied considerably over time and seasons. As an explanation of their findings, they propose the so-called microbial loop and viral shunt. The microbial loop designates the consumption of the dissolved organic matter by microorganisms and their transfer to the higher trophic layers, while the viral shunt designates the return of the carbon into the pool of bacterial substrates by the lysis of the bacteria by bacteriophages. Although the study of Evans et al. is a seasonal one, such processes could be expected as results of long-term climate changes. 

One of the hypotheses of how global warming could affect the microbial communities is stated by Danovaro et al. [154]. While studying the evolution of the archaeal communities, the authors report that the primary production is expected to decrease at tropical and mid-latitudes, while an increase is expected at high latitudes. However, some of the findings of the effects of global warming and climatic changes are contradictory, and further studies are needed [155].

## 3. Conclusions

The ecosystems of Antarctica can be viewed as highly isolated from human influence and therefore provide a unique opportunity to study and understand the impact of anthropogenic factors on the pristine microbial communities. A glimpse into how the Antarctic ecosystems respond to intense stimuli can be obtained by studies where a controlled influx of organic compounds and moisture is introduced [47,63]. For example, a gradual increase in soil temperature on an annual scale was monitored [42], or human activity in research stations and Antarctic places of interest was simply measured [140]. Indeed, these studies have shown that microbial communities in Antarctica have a low resistance to changes, and they quickly adapt to new environmental conditions by an overall decrease in diversity and by an increase in the number of generalist species that are better adapted to copiotrophic conditions. The rate at which the altered microbial communities return to their natural profile has not yet been elucidated. Still, it is safe to assume that it would take longer than that of the more temperate climate communities. This is due to the low temperatures, which inhibit growth, and the lack of dispersive forces. The wind is the most reliable method of distribution, but it greatly depends on the geography of a particular area [142]. Importantly, caution should be taken when extrapolating data from microbial communities to those of macrofauna and flora. Yet, bacteria and fungi are the pillars of every ecosystem, responsible for the crucial carbon and nitrogen cycling within them. In this sense, the impact of global warming is undeniable, with an expected 25% decrease in Antarctic ice-free areas by the end of the century [157].

## Figures and Tables

**Figure 1 life-12-00916-f001:**
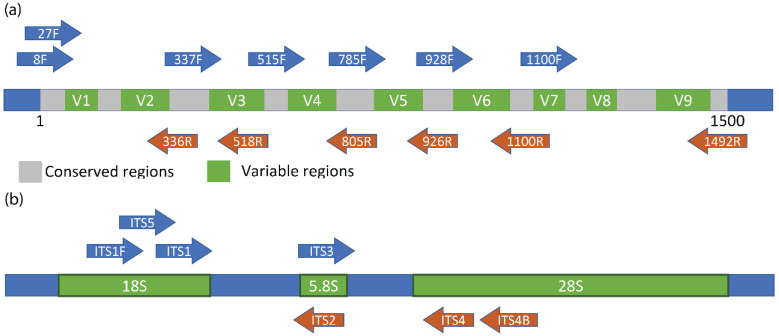
Gene layout of the two commonly used reporter molecules in NGS and some of the typical primers employed in their sequencing. (**a**) The 16S rRNA gene, used for sequencing of bacterial and archaeal communities. The variable regions allow for the delineation between OTUs, while the conservative regions are used as universal primer binding sites. (**b**) The ITS regions flanking the 5.8S rRNA gene used for annotating eukaryotes.

**Figure 2 life-12-00916-f002:**
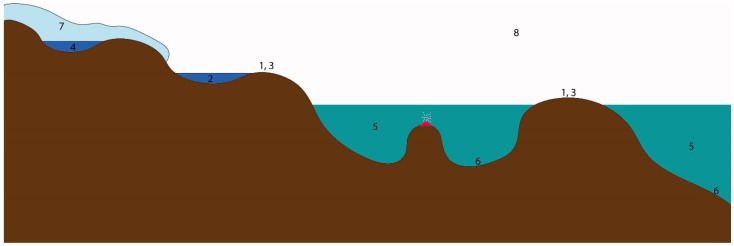
Ecological niches occupied by the different microbial communities in Antarctica. 1—Soil and rock communities; 2—Inland and glacial water communities; 3—Meltwater ponds communities; 4—Subglacial water communities; 5—Marine water communities; 6—Microbial mats and sediments communities; 7—Cryoconite hole, glacial ice, and snow communities; 8—Airborne microorganisms.

**Table 1 life-12-00916-t001:** Comparison between amplicon and shotgun sequencing.

	Amplicon Sequencing	Shotgun Sequencing
Advantages	– Can work with small sample volumes	– Very high resolution (up to SNPs)
	– Results are focused on a single kingdom	– Results for all DNA in the sample
	– Cheap	
Disadvantages	– Smaller taxonomic resolution	– Requires larger sample volumes
	– Results restricted to a single kingdom	– Results for a single kingdom are harder to distinguish
	– Variations in the number of amplicons as a result of PCR amplification or simply the nature of the organism	– Expensive for the time being

## Data Availability

Not applicable.

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
