# Peer review of "Microbial Community Composition of the Antarctic Ecosystems: Review of the Bacteria, Fungi, and Archaea Identified through an NGS-Based Metagenomics Approach"

_life, 2022, doi:10.3390/life12060916_

Round 1
Reviewer 1 Report
This paper entitled"Microbial community composition of the Antarctic ecosystems: Review of the bacteria, fungi, and archaea identified through an NGS-based metagenomics approach" is appeared to be a nice piece of work and will provide more information and reference for future study. However, I think this need major revision to accept. The main problem was most of the content is the description of previous studies, and I would like to suggest the author to summarize and refine. Moreover, I strongly recommmend the author to add the shortcomings for the microbial community composition of the Antarctic ecosystems, and the suggestions for further studies.
Author Response
Dear reviewer,
We would like to thank you for the high esteem of our work. However, as our paper is a review, and not an experimental one, we relay on the analyses of the reported on the literature results. We believe that we refined our summarizations by adding several new paragraphs. Additionally, a special paragraph including on the shortcomings of the NGS-based technology is included.
Sincerely,
The authors
Reviewer 2 Report
Review of the manuscript by Doytchinov and Dimov, entitled “Microbial community composition of the Antarctic ecosystems: Review of the bacteria, fungi, and archaea identified through an NGS-based metagenomics approach”, submitted to Life for consideration.
I commend the Authors for this very comprehensive and useful work, spanning many different Antarctic environments and types of microbes. I provide here below some hints to further improve this nice work.
- Since the abstract introduces the important aspect of biologically-active and useful products from Antarctic microbes, a specific paragraph or at least mention to some relevant literature on this should be included. See for example:
Duarte, A. W. F., Dos Santos, J. A., Vianna, M. V., Vieira, J. M. F., Mallagutti, V. H., Inforsato, F. J., ... & Durães Sette, L. (2018). Cold-adapted enzymes produced by fungi from terrestrial and marine Antarctic environments. Critical reviews in biotechnology, 38(4), 600-619.
Varrella, S., Barone, G., Tangherlini, M., Rastelli, E., Dell’Anno, A., & Corinaldesi, C. (2021). Diversity, ecological role and biotechnological potential of antarctic marine fungi. Journal of Fungi, 7(5), 391.
Silva, T. R. E., Silva, L. C. F., de Queiroz, A. C., Alexandre Moreira, M. S., de Carvalho Fraga, C. A., de Menezes, G. C. A., ... & Duarte, A. W. F. (2021). Pigments from Antarctic bacteria and their biotechnological applications. Critical Reviews in Biotechnology, 41(6), 809-826.
Núñez-Montero, K., & Barrientos, L. (2018). Advances in Antarctic research for antimicrobial discovery: a comprehensive narrative review of bacteria from Antarctic environments as potential sources of novel antibiotic compounds against human pathogens and microorganisms of industrial importance. Antibiotics, 7(4), 90.
- Similarly, in the abstract the Authors introduce the issue of climate change, but this aspect is only scarcely discussed in the review. How are the different environments changing due to climate change, and what is the possible discussion related to microbial interplay with climate change? I guess that something also about viruses should be mentioned in this review, as they have strong influence on their microbial hosts and are thus important part of the story here. See some papers on these aspects, for example
Laffoley, D., & Baxter, J. M. (Eds.). (2016). Explaining ocean warming: Causes, scale, effects and consequences. Gland, Switzerland: IUCN.
Kim, D., Park, H. J., Kim, J. H., Youn, U. J., Yang, Y. H., Casanova‐Katny, A., ... & Hong, S. G. (2018). Passive warming effect on soil microbial community and humic substance degradation in maritime Antarctic region. Journal of Basic Microbiology, 58(6), 513-522.
Rosa, L. H., Zani, C. L., Cantrell, C. L., Duke, S. O., Dijck, P. V., Desideri, A., & Rosa, C. A. (2019). Fungi in Antarctica: diversity, ecology, effects of climate change, and bioprospection for bioactive compounds. In Fungi of Antarctica (pp. 1-17). Springer, Cham.
Evans, C., Brandsma, J., Meredith, M. P., Thomas, D. N., Venables, H. J., Pond, D. W., & Brussaard, C. P. (2021). Shift from carbon flow through the microbial loop to the viral shunt in coastal Antarctic waters during austral summer. Microorganisms, 9(2), 460.
Danovaro, R., Corinaldesi, C., Dell’Anno, A., & Rastelli, E. (2017). Potential impact of global climate change on benthic deep-sea microbes. FEMS microbiology letters, 364(23), fnx214.
Danovaro, R., Rastelli, E., Corinaldesi, C., Tangherlini, M., & Dell'Anno, A. (2017). Marine archaea and archaeal viruses under global change. F1000Research, 6.
Cowan, D. A., Makhalanyane, T. P., Dennis, P. G., & Hopkins, D. W. (2014). Microbial ecology and biogeochemistry of continental Antarctic soils. Frontiers in Microbiology, 5, 154.
- Throughout the paper, please pay attention to the correct use of singular and plural when using latin words such as phylum/phyla, taxon/taxa and similar
- Throughout the paper, please rephrase each paragraph title, to better inform on which paragraph refers to terrestrial/marine/freshwater/air environment.
- Only one very simple table and one (not so informative) figure are shown, which is too few. The Authors should put additional effort in producing at least one synthetic figure, which shows the main patterns discussed in the different environments discussed in their review. This is probably the comment that will require more effort, but I think this is essential. See for inspiration for example figures in Boetius et al. NatRevMicrobiol 2015 Microbial ecology of the cryosphere: sea ice and glacial habitats, 13, 677-690.
Author Response
Dear reviewer,
We would like to thank you for the high esteem of our paper, as well for the detailed analysis and the recommendations he made.
- Since the abstract introduces the important aspect of biologically-active and useful products from Antarctic microbes, a specific paragraph or at least mention to some relevant literature on this should be included. See for example:
uarte, A. W. F., Dos Santos, J. A., Vianna, M. V., Vieira, J. M. F., Mallagutti, V. H., Inforsato, F. J., ... & Durães Sette, L. (2018). Cold-adapted enzymes produced by fungi from terrestrial and marine Antarctic environments. Critical reviews in biotechnology, 38(4), 600-619.
Varrella, S., Barone, G., Tangherlini, M., Rastelli, E., Dell’Anno, A., & Corinaldesi, C. (2021). Diversity, ecological role and biotechnological potential of antarctic marine fungi. Journal of Fungi, 7(5), 391.
ilva, T. R. E., Silva, L. C. F., de Queiroz, A. C., Alexandre Moreira, M. S., de Carvalho Fraga, C. A., de Menezes, G. C. A., ... & Duarte, A. W. F. (2021). Pigments from Antarctic bacteria and their biotechnological applications. Critical Reviews in Biotechnology, 41(6), 809-826.
Núñez-Montero, K., & Barrientos, L. (2018). Advances in Antarctic research for antimicrobial discovery: a comprehensive narrative review of bacteria from Antarctic environments as potential sources of novel antibiotic compounds against human pathogens and microorganisms of industrial importance. Antibiotics, 7(4), 90.
- We would like to thank the reviewer for this suggestion. We included such a paragraph accordingly, and we believe it increased the quality of our paper.
- Similarly, in the abstract the Authors introduce the issue of climate change, but this aspect is only scarcely discussed in the review. How are the different environments changing due to climate change, and what is the possible discussion related to microbial interplay with climate change? I guess that something also about viruses should be mentioned in this review, as they have strong influence on their microbial hosts and are thus important part of the story here. See some papers on these aspects, for example
Laffoley, D., & Baxter, J. M. (Eds.). (2016). Explaining ocean warming: Causes, scale, effects and consequences. Gland, Switzerland: IUCN.
Kim, D., Park, H. J., Kim, J. H., Youn, U. J., Yang, Y. H., Casanova‐Katny, A., ... & Hong, S. G. (2018). Passive warming effect on soil microbial community and humic substance degradation in maritime Antarctic region. Journal of Basic Microbiology, 58(6), 513-522.
Rosa, L. H., Zani, C. L., Cantrell, C. L., Duke, S. O., Dijck, P. V., Desideri, A., & Rosa, C. A. (2019). Fungi in Antarctica: diversity, ecology, effects of climate change, and bioprospection for bioactive compounds. In Fungi of Antarctica (pp. 1-17). Springer, Cham.
Evans, C., Brandsma, J., Meredith, M. P., Thomas, D. N., Venables, H. J., Pond, D. W., & Brussaard, C. P. (2021). Shift from carbon flow through the microbial loop to the viral shunt in coastal Antarctic waters during austral summer. Microorganisms, 9(2), 460.
Danovaro, R., Corinaldesi, C., Dell’Anno, A., & Rastelli, E. (2017). Potential impact of global climate change on benthic deep-sea microbes. FEMS microbiology letters, 364(23), fnx214.
Danovaro, R., Rastelli, E., Corinaldesi, C., Tangherlini, M., & Dell'Anno, A. (2017). Marine archaea and archaeal viruses under global change. F1000Research, 6.
Cowan, D. A., Makhalanyane, T. P., Dennis, P. G., & Hopkins, D. W. (2014). Microbial ecology and biogeochemistry of continental Antarctic soils. Frontiers in Microbiology, 5, 154.
- We would like to thank the reviewer for this suggestion. We included such a paragraph accordingly, and we believe it increased the quality of our paper.
Throughout the paper, please pay attention to the correct use of singular and plural when using latin words such as phylum/phyla, taxon/taxa and similar
- We are very thankful for pointing out these mistakes which were corrected.
- Throughout the paper, please rephrase each paragraph title, to better inform on which paragraph refers to terrestrial/marine/freshwater/air environment.
- We partially agree with this suggestion and changed one of the headings to “Terrestrial communities”. However, we cannot include a “Freshwater communities” heading because the structure of the text is divided into Inland and Marine communities, and also because salted waters are also present inland.
- Only one very simple table and one (not so informative) figure are shown, which is too few. The Authors should put additional effort in producing at least one synthetic figure, which shows the main patterns discussed in the different environments discussed in their review. This is probably the comment that will require more effort, but I think this is essential. See for inspiration for example figures in Boetius et al. NatRevMicrobiol 2015 Microbial ecology of the cryosphere: sea ice and glacial habitats, 13, 677-690.
- We included such a figure.
Sincerely,
The authors
Round 2
Reviewer 1 Report
The paper has been improved after revision. However, I think this need minor revision to accept. The main problem was the reference. such as reference 2, 8, 24, 32, 35 need the name of journal. I strongly recommend the authors to check the references carefully according to the requirements of the journal.
Author Response
Dear reviewer,
I would like to apologize for not remarking that the EndNote citations files were incomlete. I corrected citations 2, 8, 24 and 35 as you suggested.
Thank you for your cooperation!